# Targeting GABA_C_ Receptors Improves Post-Stroke Motor Recovery

**DOI:** 10.3390/brainsci11030315

**Published:** 2021-03-02

**Authors:** Petra S. van Nieuwenhuijzen, Kim Parker, Vivian Liao, Josh Houlton, Hye-Lim Kim, Graham A. R. Johnston, Jane R. Hanrahan, Mary Chebib, Andrew N. Clarkson

**Affiliations:** 1Brain and Mind Centre, School of Pharmacy, Faculty of Medicine and Health, The University of Sydney, Camperdown, NSW 2050, Australia; petra.s.van@gmail.com (P.S.v.N.); vivian.liao@sydney.edu.au (V.L.); hyelim.kim@gmail.com (H.-L.K.); jane.hanrahan@sydney.edu.au (J.R.H.); mary.collins@sydney.edu.au (M.C.); 2Department of Anatomy, Brain Health Research Centre and Brain Research New Zealand, University of Otago, Dunedin 9054, New Zealand; kim.parker@otago.ac.nz (K.P.); josh.houlton@otago.ac.nz (J.H.); 3Discipline of Pharmacology, Faculty of Medicine and Health, The University of Sydney, Sydney, NSW 2006, Australia; graham.johnston@sydney.edu.au; 4Department of Anatomy, University of Otago, Dunedin 9054, New Zealand

**Keywords:** photothrombotic stroke, tonic inhibition, motor behavior, GABA, GABA receptors, reactive astrogliosis

## Abstract

Ischemic stroke remains a leading cause of disability worldwide, with limited treatment options available. This study investigates GABA_C_ receptors as novel pharmacological targets for stroke recovery. The expression of *ρ*1 and **ρ**2 mRNA in mice were determined in peri-infarct tissue following photothrombotic motor cortex stroke. (*R*)-4-amino-cyclopent-1-enyl butylphosphinic acid (*R*)-4-ACPBPA and (*S*)-4-ACPBPA were assessed using 2-elecotrode voltage electrophysiology in *Xenopus laevis* oocytes. Stroke mice were treated for 4 weeks with either vehicle, the α5-selective negative allosteric modulator, L655,708, or the *ρ*1/2 antagonists, (*R*)-4-ACPBPA and (*S*)-4-ACPBPA respectively from 3 days post-stroke. Infarct size and expression levels of GAT3 and reactive astrogliosis were determined using histochemistry and immunohistochemistry respectively, and motor function was assessed using both the grid-walking and cylinder tasks. After stroke, significant increases in *ρ*1 and *ρ*2 mRNAs were observed on day 3, with *ρ*2 showing a further increase on day 7. (*R*)- and (*S*)-4-ACPBPA are both potent antagonists at *ρ*2 and only weak inhibitors of α5β2γ2 receptors. Treatment with either L655,708, (*S*)-4-ACPBPA (*ρ*1/2 antagonist; 5 mM only), or (*R*)-4-ACPBPA (*ρ*2 antagonist; 2.5 and 5 mM) from 3 days after stroke resulted in a significant improvement in motor recovery on the grid-walking task, with L655,708 and (*R*)-4-ACPBPA also showing an improvement in the cylinder task. Infarct size was unaffected, and only (*R*)-4-ACPBPA significantly increased peri-infarct GAT3 expression and decreased the level of reactive astrogliosis. Importantly, inhibiting GABA_C_ receptors affords significant improvement in motor function after stroke. Targeting the *ρ*-subunit could provide a novel delayed treatment option for stroke recovery.

## 1. Introduction

Ischemic stroke is one of the most common causes of severe disability worldwide [1]. To date, few treatments options are available, and those that are available such as clot busters are only effective if given within a narrow therapeutic time window (4.5–6 h) after stroke onset [2]. As a consequence, only 5–10% of stroke survivors will receive thrombolysis, leaving the remaining 90–95% dependent on physical therapy to improve their recovery outcomes. Stroke survivors often experience lasting motor impairments; therefore, there is an urgent need to find novel therapies that can be administered days to weeks after stroke to help improve motor function.

Changes in γ-aminobutyric acid (GABA) function following cerebral ischemia and the protective benefits of GABAergic compounds targeting synaptic GABA_A_ receptors have been reported [3,4,5,6]. Despite GABA_A_ receptor agonists showing great promise in animal models of stroke, they have failed to translate into positive clinical outcomes [7], which is most likely due to a lack of subtype specificity resulting in unwanted side effects, hence limiting their therapeutic potential [8,9,10].

Over the past decade, preclinical research has highlighted several mechanisms related to post-stroke recovery [11,12,13,14]. Many of these processes involve changes in molecular signaling pathways, which are closely linked to changes in learning and memory [15] and an imbalance in brain excitability [5,16,17].

By selecting compounds and dosing paradigms that selectively target individual GABA receptor subtypes, researchers have shown that the modulation of either tonic or phasic inhibition can enhance functional motor recovery in rodents after stroke [5,18,19]. GABA_C_ receptors also known as GABA_A_ rho (*ρ*) receptors are insensitive to bicuculline; they can be found extrasynaptically, and similar to other extrasynaptic GABA_A_ receptors (e.g., α5- or δ-containing GABA_A_ receptors), they do not readily desensitize in the presence of GABA [20,21]. In addition to modulating tonic inhibitory currents directly by targeting extrasynaptic GABA_A_ receptors, we have recently also reported that targeting the astrocytic GABA transporter type 3 (GAT3) [22] using the substrate, L-isoserine, increases the expression of GAT3 and improves motor function [19].

Given the extrasynaptic location of GABA_C_ receptors, and as their expression are in regions involved in motor control such as the neostriatum and cerebellum [23,24], we envisaged that such receptors could be an interesting target to explore for stroke recovery. As GABA_C_ receptors are expressed on both neurons and astrocytes [25,26,27], we asked the question whether inhibiting GABA_C_Rs can also play a role in improving motor function.

(*R*)- and (*S*)-4-amino-cyclopent-1-enyl butylphosphinic acid ((*R)*-4-ACPBPA and (*S*)-4-ACPBPA) are compounds that selectively inhibit the *ρ*-subunits of GABA_C_ receptors [22,28]. Both compounds are inactive at the α1β2γ2 GABA_A_ receptors when tested at doses < 600 µM and inactive at the GABA_B_ receptor at doses < 300 µM [28]. Furthermore, both compounds have recently been evaluated in vivo and are efficacious at preventing ethanol-induced motor incoordination in mice [22].

In the present study, we used (*R*)-4-ACPBPA and (*S*)-4-ACPBPA to target GABA_C_ receptors and measured changes in functional recovery, and glial markers, glial fibrillary acidic protein (GFAP), and GAT3 expression after inducing photothrombotic stroke. These studies were compared to the α5-subunit negative allosteric modulator (NAM), L655,708, which improves motor recovery preclinically after stroke [5].

## 2. Materials and Methods

### 2.1. Animals

All procedures described in this study were approved by the University of Otago, Animal Ethics Committee in accordance with the ARRIVE (Animal Research: Reporting of In Vivo Experiments) guidelines (AEC #92/11). As GABA_C_R modulators have not been tested before in a stroke model, it is widely accepted that all testing be undertaken in young male mice first. A total of 130 young (2–3-month-old) male C57BL/6*J* mice were used for these studies, which allows us to compare and validate results to previous published studies. All mice were housed under a 12 h light/12 h dark cycle (lights on 0700 h) with ad libitum access to food and water. All mice were group housed (*n* = 3–5/cage) in Tecniplast caging with corncob bedding. Cages were enriched with shredded paper, plastic tunnels, and wood blocks for chewing, with new enrichments replaced weekly. All mice were assigned to either stroke or sham experimental groups at the time of surgery by one member of staff, with both groups being further randomly assigned to a treatment group 3 days later by a second member of staff to ensure all studies were undertaken in a blinded fashion. All samples sizes for the various assessment parameters were calculated based on our own prior studies and the use of sample size calculators [5,29].

### 2.2. Allocation of Animals to Different Experiments

For the qPCR experiment, a total of 30 animals, 5 for each time point (0, 1, 3, 7, 14, and 28 days post-stroke) were used. For behavioral assessments, a total of 100 animals were divided into 10 different groups of 10 animals: (1) Sham + Vehicle, (2) Sham + L655,708, (3) Sham + 5 mM (*S*)-4-ACPBPA, (4) Sham + 5 mM (*R*)-4-ACPBPA, (5) Stroke + Vehicle, (6) Stroke + L655,708, (7) Stroke + 2.5 mM (*S*)-4-ACPBPA, (8) Stroke + 5 mM (*S*)-4-ACPBPA, (9) Stroke + 2.5 mM (*R*)-4-ACPBPA, (10) Stroke + 5 mM (*R*)-4-ACPBPA. Of these animals, six from each group were randomly selected to determine infarct size, GAT3, and GFAP analysis. For the combinatorial studies, a total of 48 animals were divided into 6 different groups of 8 animals: (1) Sham + Vehicle, (2) Sham + 5 mM L655,708 + 5 mM (*R*)-4-ACPBPA, (3) Stroke + Vehicle, (4) Stroke + L655,708, (5) Stroke + 5 mM (*R*)-4-ACPBPA, (6) Stroke + 5 mM L655,708 + 5 mM (*R*)-4-ACPBPA.

### 2.3. Photothrombosis Model of Focal Ischemia

Focal stroke was induced in the left hemisphere using the photothrombosis method in young adult (2–3 month old, 27–30 g) male C57BL/6*J* mice [5,29]. In brief, under isoflurane anesthesia (2–2.5% in medical O_2_), mice were placed in a stereotaxic apparatus, the skull exposed and a cold light source (KL1500 LCD, Carl Zeiss, Auckland, New Zealand) attached to a 20× objective to give a 2 mm diameter illumination positioned 1.5 mm lateral from Bregma. Animals were administered 0.2 mL of Rose Bengal solution (Sigma-Aldrich, Auckland, New Zealand; 10 g/L in normal saline) intraperitoneally (i.p.) and left for five minutes to allow circulation prior to a 15 min illumination of the brain. Body temperature was maintained at 36.9 ± 0.3 °C with a heating pad (Harvard apparatus, Holliston, MA, USA) throughout surgical procedures. Following the light exposure, the skin was closed using surgical glue. After a brief recovery period, animals were returned to their normal housing conditions. Sham animals received saline instead of Rose Bengal solution.

Animals were randomly allocated to different treatment groups: L655,708 (5 mM pump concentration), (*R*)-4-ACPBPA (2.5 mM and 5 mM pump concentrations), (*S*)-4-ACPBPA (2.5 mM and 5 mM pump concentrations), combined L655,708 and (*R*)-4-ACPBPA (5 mM pump concentration each) or vehicle. Drugs were dissolved in DMSO and then diluted 1:1 in 0.9% saline. Vehicle-treated animals were implanted with pumps filled with 1:1 DMSO and 0.9% saline solution. All compounds have previously been tested in vivo and shown to cross the BBB and have an effect on the brain, including L655,708, which we have previously reported is effective after stroke [5].

Drug or vehicle-filled ALZET-1002 pumps (DURECT Corporation: Cupertino, CA, USA) were implanted on day 3 post-stroke and replaced every two weeks [5]. The concentration in one minipump, 5 mM, delivers a 200 µg/kg/day dose in mice. To implant the ALZET-1002 pumps, mice were anesthetized with isoflurane/O_2_ mixture, and a small (<1 cm) incision was made between the shoulder blades of the animals. A subcutaneous pocket was made by gently inserting blunt forceps under skin. The pump was placed with port side facing away from the incision and the incision closed using surgical glue. During all surgical procedures, mice received temgesic as pain relief on the day of surgery as well as the following day. In addition, for minipump implantation procedures, topical lidocaine was also given.

To ensure that all experiments were undertaken in a blind manner, animals were allocated into each of the treatment groups, and minipumps were implanted by someone not undertaking any of the assessments (behavior, real-time qPCR, immunohistochemical, and histological stains). For this, animals in each cage were assigned a number, and then each number was randomly assigned to a treatment.

### 2.4. Tissue Collection

All mice at the end of the study were sacrificed by anesthetic overdose with pentobarbital (100 mg/kg). For tissue being collected for qPCR analysis, brains were rapidly removed from the cranium, frozen on dry ice, and stored at −80 °C prior to being used. All other mice underwent transcardiac perfusion with 4% formaldehyde, and brains were removed for histochemical assessments.

### 2.5. Real-Time qPCR Measurement of GABA Subunits (n = 5 at Each Time Point)

The relative mRNA expression of GABA subunits (α5, *ρ*1 and *ρ*2) were assessed by quantitative real-time polymerase chain reaction (qPCR) [17] in *n* = 5 mice at each time point. Bain tissue samples that included both the stroke and surrounding peri-infarct area were collected and snap-frozen at 0 °C (control non-stroked tissue), 1, 3, 7, 14, or 28-days post-stroke. Total RNA was extracted using the Qiagen RNeasy kit and following the manufacturer’s protocols. The purity (RNA with ratio of absorbance at 260 nm and 280 nm ≥ 2) and amount of the RNA was measured spectrophotometrically (NanoDrop 2000, Thermo Scientific, Waltham, MA, USA). Total RNA (750 ng) was used to synthesize the first-strand complementary DNA (cDNA) using Super Script III (Life Technologies, Waltham, MA, USA) following the manufacturer’s protocol. After reverse transcription, the cDNA was amplified by qPCR using SyBr green master mix (Applied Biosystems, Foster City, USA) and each of the following primer (250 nM) sets; α5: forward—GCTGACCCATCCTCCAAACA, reverse—TGGAGACTGTGGGTGCATTC; *ρ*1: forward—CTTCTCACGGCTTCTTGGGA, reverse—ACCCATCCCCACCACAAAAG; *ρ*2: forward—CCATTAAAAGTCCCTGCACAGC, reverse—ATGTTTCCAGAAGCCCTGTCC; Rpl13a: forward—ATTGTGGCCAAGCAGGTACT, reverse—CTCGGGAGGGGTTGGTATTC. qPCR was performed on a Roche Lightcycler 480 (Roche, Minneapolis, USA) with the following cycling parameters: 40 cycles of 95 °C, 15 s; 60 °C, 30 s; 72 °C, 40 s. After amplification, a denaturing curve was performed to ensure the presence of unique amplification products. All reactions were performed in triplicate. Expression of mRNA was assessed by evaluating threshold cycle (CT) values. The CT values were normalized against the expression level of the house keeping gene succinate dehydrogenase complex flavoprotein complex A (SDHA).

### 2.6. Electrophysiology in Xenopus laevis Oocytes

All procedures using *Xenopus laevis* were approved by the animal ethics committee of the University of Sydney (AEC No. 2013/5269). Female *Xenopus laevis* of approximately 1 year of age were from Xenopus Express Brookville, FL, USA. To obtain isolated oocytes, ovarian lobes were removed from anesthetized adult female frogs incised into small pieces using surgical scissors and defolliculated by collagenase A treatment. Stage V and VI oocytes were injected with cRNA mixture encoding α5, β2, and γ2 at a ratio of 5:1:5, 25 ng/cell, or 50 ng/cell *ρ*2 cRNA GABA_A_R subunits and were incubated at 18 °C for 3–5 days in ND96 solution (96 mM NaCl, 2.0 mM KCl, 1 mM MgCl2, 1.8 mM CaCl_2_, 5 mM HEPES, 2.5 mM sodium pyruvate, 0.5 mM theophylline, and 50 µg/mL gentamicin; pH 7.4) before electrophysiological experiments. Stock solutions of GABA (1 mM), (*S*)-4-ACPBPA (100 mM), and (*R*)-4-ACPBPA (100 mM) were prepared by dissolving solid compounds in ultrapure water.

Whole-cell currents were recorded using the two-electrode voltage clamp technique. To determine the GABA inhibitory activity of (*S*)-4-ACPBPA and (*R*)-4-ACPBPA on *ρ*2 receptors, inhibitory concentration response curves of (*S*)-4-ACPBPA and (*R*)-4-ACPBPA were co-applied with GABA 1 µM (≈EC_50_) with oocytes clamped at −60 mV. Data were acquired with a LabChart v 3.5.2 (ADInstruments Pty Ltd, New South Wales, Australia) analogue to digital converter, and currents low-pass-filtered at 1 kHz and sampled at 3 kHz were measured offline with LabChart v3.5.2 software. The bath solution contained ND96 solution, and electrodes were filled with 3M KCl (0.5–2 MΩ). Solutions were bath applied using a gravity-fed perfusion system.

To determine the GABA inhibitory activity of (*S*)-4-ACPBPA and (*R*)-4-ACPBPA on α5β2γ2 receptors, two concentrations of (*S*)-4-ACPBPA and (*R*)-4-ACPBPA, 10 µM and 100 µM, were co-applied with GABA 10 µM (≈EC_30_). To ensure the reproducibility of the evoked current amplitudes, a set of control applications was performed before the inhibitory experiments: three GABA_control_ (10 µM) applications, one GABA_max_ (316 µM approximately EC_100_) application, and another three GABA_control_ (10 µM) applications. This is followed by the co-application of GABA _control_ (10 µM) with (*S*)-4-ACPBPA (10 µM) or (*R*)-4-ACPBPA (10 µM), then GABA _control_ (10 µM) to ensure the current return to previous control level, and then the co-application of GABA _control_ (10 µM) with (*S*)-4-ACPBPA (100 µM) or (*R*)-4-ACPBPA (100 µM). Data were assembled from a minimum of 6 independent experiments (*n* = 6) with errors expressed as SEM.

### 2.7. Behavioral Assessment (n = 8–10/Group)

Animals were tested one week prior to surgery on both the grid-walking and cylinder tasks to establish baseline performance levels. Then, all animals were tested on weeks 1, 2, 3, 4, and 6 weeks post-stroke at approximately the same time each day, at the end of their dark cycle. All behaviors were scored by observers who were blind to the treatment group of the animals in the study as previously described [5,16]. Ten animals per group were assessed on all behavioral tasks, except for the combinatorial studies, where we used an *n* = 8/group.

### 2.8. Grid-Walking Test

The grid-walking apparatus was manufactured using 12 mm square wire mesh with a grid area 32 cm/20 cm/50 cm (length/width/height). A mirror is placed beneath the apparatus to allow video footage in order to assess the animals’ stepping errors (i.e., “foot-faults”). Each mouse is placed individually atop of the elevated wire grid and allowed to freely walk for a period of 5 min (measured in real time by stopwatch and confirmed afterwards by reviewing videotape footage). During this 5-min period, the total number of foot-faults for each limb along with the total number of non-foot-fault steps are counted, and a ratio between foot-faults and total steps taken was calculated. The percent of foot-faults was calculated as follows: (#foot-faults/(#foot-faults + #non-foot-fault steps) * 100). A ratio between foot-faults and total steps taken was used to take into account differences in the degree of locomotion between animals and trials.

### 2.9. Cylinder Task

The spontaneous forelimb task encourages the use of forelimbs for vertical wall exploration/press in a cylinder. When placed in a cylinder, the animal rears to a standing position, whilst supporting its weight with either one or both of its forelimbs on the side of the cylinder wall. A cylinder 15 cm in height with a diameter of 10 cm is used. Videotape footage of animals in the cylinder is evaluated quantitatively in order to determine forelimb preference during vertical exploratory movements. While the video footage is played in slow motion (1/5th real time speed), the time (s) during each rear that each animal spent on either the right forelimb, the left forelimb, or on both forelimbs are calculated. Only rears in which both forelimbs could be clearly seen are timed. From these three measures, the total amount of time spent on either limb independently as well as the time the animal spent rearing using both limbs was derived. The percentage of time spent on each limb was calculated, and these data were used to derive a spontaneous forelimb asymmetry index (% ipsilateral use/% contralateral use). The “contact time” method of examining the behavior was chosen over the “contact placement” method, as it takes into account the slips that often occur during a bilateral wall press post-stroke.

### 2.10. Immunohistochemical and Histological Assessments (n = 6)

Following the final behavioral assessment at 6 weeks post stroke, all animals were anesthetized and transcardially perfused with 4% paraformaldehyde. Brains were extracted, and 30 μm thick coronal sections were collected using a sliding microtome. Then, tissue was processed histologically using cresyl violet staining in order to quantify infarct volume as previously described [5,16,19,30]. Images of cresyl violet staining were taken using an inverted montaging microscope (Model Ti2E Wideview, Nikon, Japan) set with a 2.5× objective lens. Then, images were exported as TIFF files and opened on Fiji ImageJ to quantify infarct volume. Stroke volume was calculated as per the equation:

$$Infarct\;Volume\;\left(mm^{3}\right)=Area\;\left(mm^{2}\right)\;\times \;Section\;Thickness\;\times \;Section\;Interval$$

Immunofluorescent labeling of GAT3 and GFAP was performed 42 days post stroke to assess the effects of GABA treatments on glial scar formation and the astrocytic GABA transporter, respectively. Briefly, sections were washed thoroughly in Tris-buffered saline (TBS), blocked in 5% donkey serum, and incubated for 48 h at 4 °C in either the polyclonal chicken anti-GFAP (1:2000; Millipore, Burlington, MA, USA) or in the rabbit anti-mouse GAT3 (1:100, Millipore, Burlington, MA, USA) primary antibodies, which were diluted in TBS containing 0.3% Triton X-100 and 0.25% bovine serum albumin (hereafter referred to as incubation solution) containing 2% normal donkey serum. Then, sections were washed three times in TBS (10 min per wash) before being incubated in either the donkey anti-chicken 549 secondary antibody (1:400; Jackson Immunoresearch, West Grove, PA, USA) or the donkey anti-mouse 488 DyLight (1:400; Sapphire Bioscience, New South Wales, Australia) secondary antibody in incubation solution for 90 min at room temperature. After subsequent washing in TBS, sections were mounted onto gelatin-coated glass slides, air-dried, and passed sequentially through alcohols (50%, 70%, 95%, and 100%) before being passed through xylene and then cover slipped using DPX mounting solution.

Images of the glial scar (400 µm from the stroke border) encompassing what is known as the peri-infarct region were taken on an Olympus BX61 (Olympus, Auckland, New Zealand) microscope using a QImaging camera (Micropublisher 5.0 RTV, 2560 × 1920 pixels, 3.4 µm pixel size; Olympus, Auckland, New Zealand) and 20× UPlanSApo objective (0.75NA: Olympus, Auckland, New Zealand). Fluorescent intensity measures were taken using ImageJ (National Institutes of Health, Bethesda, MA, USA) analysis. Fluorescent intensity measurements of the scar were normalized to background reading on the contralateral hemisphere, and an average fluorescence was obtained using 3 sections per animal (*n* = 6). Images and fluorescent intensity measurements were conducted by observers blinded to treatment group allocation.

### 2.11. Statistical Analysis

All statistical analyses were performed using GraphPad Prism (version 7.0c, San Diego, CA, USA). One- or two-way analysis of variance (ANOVA) and Tukey’s multiple pair-wise comparisons for post hoc comparisons were used. Electrophysiological data are represented as mean ± SEM, all other data as mean ± SD, and *p* < 0.05 was considered statistically significant.

### 2.12. Power Analysis

For behavioral experiments, 6 animals per group are required to achieve >80% power (86% calculated), considering the following parameters: α = 0.05; with an effect size = 1.5. For qPCR and histological/immunohistochemical experiments, 5 animals per group are required to achieve >80% power (91% calculated), considering the following parameters: α = 0.05; effect size 1; 3 concentrations; 2 groups, and correlation between measures =0.5. Parameters were determined from our prior work, in which we have demonstrated significant behavioral effects [5,17,19,31], and on the assumption that variance was about 25%. It should be noted that more conservative effect sizes were used for these experiments, as it is harder to assess recovery over time between groups than looking at the effects of drug treatments on stroke size.

## 3. Results

### 3.1. Real-Time qPCR

Studies to date have focused on assessing changes in GABA_A_ and GABA_B_ receptor subunits post-injury. As GABA_C_
*ρ*-subunits are also present in the brain, we aimed to assess changes in *ρ*1 and *ρ*2-subunit expression and compare this to changes in the GABA_A_ α5 subunit across time in a mouse model of focal ischemic stroke. The rational for comparing to the α5 subunit is that GABA_C_
*ρ*-subunits are also found extrasynaptically and play a role in regulating tonic inhibition. In order to assess changes in expression, cortical tissue that included stroke and peri-infarct regions only were isolated from sham (control) and stroked animals 1, 3, 7, 14, and 28 days post-insult. Comparisons between hemispheres were not undertaken, as it is well documented that the contralateral hemisphere undergoes changes after stroke to compensate for the damage and is therefore not an appropriate control.

Previous studies have reported an increase in α5-mediated tonic inhibition [5]. Consistent with these earlier findings, we report a significant increase in α5 subunit expression in tissue that encompasses the peri-infarct cortex from day 3 post stroke, with an elevation in expression still observed on day 28 post-stroke (Figure 1A). Even though *ρ*1 and *ρ*2 are expressed in the brain and found extrasynaptically, little is known about their function. Assessment of *ρ*1 and *ρ*2-subunits revealed that they were both elevated after stroke with significance observed for *ρ*1 3 days after stroke (Figure 1B) and for *ρ*2 on days 3 and 7 post-stroke (Figure 1C).

### 3.2. Electrophysiology in Xenopus Oocytes (n = 6 Independent Experiments)

Having verified that there is a change in *ρ*1 and *ρ*2-subunit expression, we assessed the role of *ρ*1 and *ρ*2-subunits using (*S*)-4-ACPBPA and (*R*)-4-APCBPA. (*S*)-4-ACPBPA and (*R*)-4-APCBPA were previously reported to be competitive antagonists at *ρ*1 [28]. Here, we show that the compounds are also competitive antagonists at *ρ*2; Figure 2A,B) as observed from the parallel rightward shift in the GABA concentration response curve with increasing concentrations of the compounds. (*S*)-4-ACPBPA is not selective for either *ρ*1 (Kd = 5 µM) [28] or *ρ*2 (10.54; Figure 2A), while (*R*)-4-ACPBPA has greater affinity for *ρ*2 receptors with (Kd = 6 µM) over *ρ*1 containing receptors (Kd =60 µM; Figure 2B) [28].

(*R*)-4-ACPBPA and (*S*)-4-ACPBPA are weak inhibitors of GABA on α5β2γ2 receptors with inhibitory activity of 2.5 ± 1.7% and 5.5 ± 2.5 % at 10 µM, 36 ± 2.0% and 37 ± 1.5% at 100 µM respectively (Figure 2C). There were no observable differences in GABA inhibitory activity between (*R*)-4-ACPBPA and (*S*)-4-ACPBPA on α5β2γ2 receptors.

### 3.3. Treatment with *ρ*-Subunit Modulators Improves Motor Recovery after Stroke (n = 10/Group)

Tonic inhibitory currents are elevated from 1 to 3 days after stroke, and modulating these currents can afford significant improvement in functional recovery [5,17]. As GABA_C_
*ρ*-subunits are located extrasynaptically [20,21] and in regions important for motor control, we aimed to assess if modulating their function could also improve functional recovery after stroke.

(*S*)-4-ACPBPA and (*R*)-4-ACPBPA were assessed for their ability to improve recovery of motor function after stroke using the grid-walking (Figure 3A and Figure 4A, respectively) and cylinder (Figure 3B and Figure 4B, respectively) tasks.

Assessment of functional recovery following delayed chronic treatment with either L655,708 or (*S*)-4-ACPBPA revealed a significant treatment effect on both the grid-walking and cylinder tasks. Post-hoc analysis revealed that treatment with 2.5 mM (*S*)-4-ACPBPA showed an improvement in motor function on days 28 and 42 post-stroke as evidenced by a 36% improvement in motor function by 42 days compared to 19% for vehicle-treated stroke controls. Treatment with 5 mM (*S*)-4-ACPBPA showed a 20% improvement in function at day 7 after stroke and reaching 38% improvement 42 days post-stroke. Treatment with (*S*)-4-ACPBPA showed no improvement in motor function as assessed in the cylinder task (Figure 3B).

Assessment of functional recovery following delayed chronic treatment with either L655,708 or (*R*)-4-ACPBPA revealed a significant treatment effect on both the grid-walking and cylinder tasks. Post-hoc analysis revealed that treatment with 2.5 mM (*R*)-4-ACPBPA showed an improvement in motor function on days 7, 14, 21, 28, and 42 post-stroke (17% and 39% on day 7 and 42 post-stroke, respectively), whilst treatment with 5 mM (*R*)-4-ACPBPA showed a marked improvement in function from day 7 after stroke (29% and 43% on day 7 and 42 post-stroke, respectively), with the level of improvement not being different than the improvement following L655,708 treatment (53% 42-days post-stroke). In addition, treatment with 5 mM (*R*)-4-ACPBPA also showed an improvement in motor function as assessed in the cylinder task on days 21 (33% improvement), 28 (36% improvement), and 42 (37% improvement) post-stroke (Figure 3B).

### 3.4. Delayed Treatment with *ρ*-subunit Antagonists Do Not Alter Infarct Size (n = 6/Group)

Consistent with what we have previously reported [5], we show that delayed administration (from 3 days post-stroke) of L655,708 (1.44 ± 0.16, *n* = 6) does not alter infarct size (Figure 5) as measured 42 days post-stroke compared to vehicle-treated stroke controls (1.62 ± 0.41, *n* = 6). Delayed treatment with *ρ*-subunit antagonists revealed no treatment effect on infarct size. Similar to what was observed following treatment with L655,708, assessment of infarct volume following treatment with either (*S*)-4-ACPBPA (1.32 ± 0.09, *n* = 6) or (*R*)-4-ACPBPA (1.33 ± 0.14, *n* = 6) revealed no difference in infarct volume as measured 42 days post-stroke compared to stroke + vehicle controls.

### 3.5. Delayed Treatment with *ρ*-Subunit Antagonists Alters Peri-Infarct GFAP and GAT3 Expression (n = 6/group)

To investigate whether any of the treatments (L655,708, (*S*)-4-ACPBPA or (*R*)-4-ACPBPA) had any effect on GAT3 expression, we assessed GAT3 expression using immunofluorescent labeling 42 days post-stroke (Figure 6A,B). Assessment of peri-infarct GAT3 levels revealed a significant treatment effect (F(3,36) = 3.348: *p* = 0.0296). Post-hoc analysis revealed that all three treatments (L655,708, (*S*)-4-ACPBPA, or (*R*)-4-ACPBPA) resulted in a significant increase in GAT3 expression compared to stroke + vehicle controls.

Lastly, we wanted to investigate whether treatment with *ρ*-subunit antagonists had any effect on peri-infarct reactive astrogliosis as assessed using GFAP immunofluorescence. Assessment of GFAP immunofluorescence 42 days post-stroke revealed extensive peri-infarct reactive astrogliosis (Figure 6A,C). Chronic drug treatment from 3 days post-stroke showed a significant treatment effect (F(3,40) = 3.491: *p* = 0.0243), although a significant decrease in reactive astrogliosis was only observed following treatment with 5 mM (*R*)-4-ACPBPA (Figure 6C) compared to stroke + vehicle controls.

### 3.6. Combined Treatment with *ρ*-Subunit Modulator, (R)-4-ACPBPA, and the GABA_A_ α5 NAM, L655,708, Markedly Improves Motor Recovery after Stroke (n = 8/Group)

Tonic inhibitory currents can be altered via a number of mechanisms, changes in GABA_A_, GABA_B_, or GABA_C_ receptor function, as well as changes in the function of astrocytic GABA transporters, GAT3 and BGT1. Given that several pathways play a role in mediating a change in tonic inhibitory currents, it is likely that a combination of treatments targeting more than one of these pathways will offer the greatest improvement in function. We have shown that treatment with either L655,708 (5 mM) with (*R*)-4-ACPBPA (5 mM) results in a decrease in the number of foot faults on the grid-walking test as well as an improvement in forelimb use in the cylinder task. As both of these compounds target different receptor subunits linked to changes in tonic inhibitory currents, we therefore undertook a combined study using L655,708 (5 mM) and (*R*)-4-ACPBPA (5 mM) to assess if a combinatorial approach would offer greater recovery post-stroke.

Mice that received the combined L655,708 and (*R*)-4-ACPBPA treatment showed a marked improvement in motor function on both the grid-walking (Figure 7A) and cylinder (Figure 7B) tasks on weeks one and two post-stroke compared to L655,708 or (*R*)-4-ACPBPA treatment alone. Assessment of motor function on weeks four and six post-stroke revealed the combined L655,708 and (*R*)-4-ACPBPA treated animals performed better than (*R*)-4-ACPBPA treatment alone (67% vs. 39% at four-weeks; and 74% vs. 49% at 6-weeks), but no difference was observed compared to L655,708 treatment alone.

## 4. Discussion

The current study investigated whether targeting GABA_C_Rs can promote motor recovery after stroke, as these receptors have previously been reported to mediate tonic inhibition and are located in brain regions important for motor control [21,27]. We show that GABA_C_-*ρ*2 relative mRNA expression increased in the peri-infarct region 3 and 7 days after stroke, with GABA_C-_*ρ*1 also showing an increase 7-days after stroke, although it is not clear whether these increases result in functional changes. Therefore, in order to test whether GABA_C_
*ρ*-containing receptors play a role in motor recovery after stroke, we evaluated the *ρ*-antagonists (*S*)-4-ACPBPA and (*R*)-4-APCBPA in a photothrombotic mouse model of stroke. L655,708, the GABA_A_ α5 NAM, was used as a positive control, as inhibiting tonic currents mediated by α5-containing receptors improves motor function from 3 days post-stroke [5,32].

All three compounds, L655,708, (*S*)-4-ACPBPA, and (*R*)-4-ACPBPA improved motor function from 3 days post stroke. On the grid-walking task, (*R*)-4-ACBPBA was more effective than the *S*-enantiomer, observing a dose-dependent improvement in performance (at 2.5 and 5 mM); however, these mice never reached the same level of performance as mice treated with L655,708. Measuring the use of the impaired limb using the cylinder task, it is clear that L655,708 was more effective than (*R*)-4-ACPBPA. The *S*-enantiomer was only effective in improving performance on the grid-walking test, and this was only observed at the highest dose tested. No significant effect was observed at the lower dose. Furthermore, (*R*)-4-ACPBPA but not (*S*)-4-ACPBPA or L655,708 decreased the expression of GFAP positive reactive astrocytes and increased GAT3 expression. In contrast, infarct volume was unchanged in all treatment groups, indicating that when treatment is started from 3 days post-stroke, the infarct is already fully formed and any behavioral improvements are the result of brain plasticity. A possible explanation for the differences in GABA_A_ and GABA_C_-mediated functional recovery is that the expression of the GABA_A_ α5-subunit is higher than that of the GABA_C_
*ρ*1 and *ρ*2-subunits. At the time of undertaking these experiments, we did not realize this was going to be an issue, as the expression of the α5-subunit is low compared to the expression of most other GABA subunits and that the receptor occupancy achieved following 5 mM L655,708 dosing via ALZET osmotic pumps is about 9–14% receptor occupancy ([5], and unpublished data). Given the relatively low receptor occupancy required to achieve a functional improvement when targeting the α5-subunit with L655,708, it is plausible that some of the effects observed with either (*R*)-4-ACPBPA or (*S*)-4-ACPBPA could be occurring via the α5-subunit. However, we believe that this is not the case, as (*R*)-4-ACPBPA was the only compound to show any glial effects.

Neither compound acts on the synaptic α1β2γ2 GABA_A_ receptor nor do they activate or inhibit GABA_B_ receptors at the doses tested [28]. However, (*S*)-4-ACPBPA and (*R*)-4-APCBPA differ in their affinity to act as an antagonist at the *ρ*2 subtype; (*R*)-4-APCBPA was more effective at inhibiting *ρ*2, and this provides some insight toward understanding the role that these subunits play in motor recovery. Both L655,708 and (*R*)-4-APCBPA afford an early improvement in motor functional recovery, which is a mechanistic process that we have only ever observed via a change in tonic inhibitory currents [5]. Given that the modulation of GABA_A_ and GABA_C_ receptors improves motor function via different mechanisms, we chose to next perform a combinatorial study, which we show resulted in a marked improvement in motor recovery that was better than L655,708 or (*R*)-4-APCBPA alone. The only difference being that the recovery trajectory in the combined group is slower across time than that of the L655,708 treatment group; however, it is possible that we are reaching the maximum possible recovery achievable in these animals. The observed synergistic effects of combined dosing with L655,708 and (*R*)-4-ACPBPA support our opinion that (*R*)-4-ACPBPA is acting on the GABA_C_
*ρ*2-subunit containing extrasynaptic receptors, as dosing with α5-subunit selective compounds that achieve either 8% or 60% receptor occupancy results in the same level of motor functional recovery (unpublished data).

The persistent elevation of tonic currents seen in the peri-infarct area after stroke is due to a lack of GAT3 function [5,19]. GAT3 is a GABA transporter located on astrocytes that removes GABA from the extrasynaptic space [33]. Without GAT3, GABA remains in the extrasynaptic space and continues to activate extrasynaptic receptors, resulting in the persistence of tonic currents. It is thought that the increase in tonic inhibition is an intrinsic mechanism that helps limit the spread of damage in the brain and silences communication so the brain can repair itself [5,17]. However, this increase in tonic inhibition is limiting the recovery of lost functions, as it remains elevated and chronically silences connections, and it becomes detrimental for stroke recovery, since it makes it harder for neurons to form novel connections and contribute to the plasticity necessary to improve motor functioning [15]. While the current study did not measure tonic currents, the increase in GAT3 expression in the peri-infarct area after treatment with 5 mM (*R*)-4-ACPBPA indicates that the functional improvements seen on grid-walking and cylinder tests may be the result of decreased tonic inhibition and enhanced neuroplasticity. This is similar to what we have previously observed following treatment with either L-isoserine [19] that resulted in an increase in GAT3 expression and improved functional recovery, and also following L655,708 treatment that dampened that stroke-induced elevation in tonic inhibition via GABA_A_ α5 receptors [5,32]. In addition to the GAT3-mediated increases in tonic inhibitory currents, previous research has also reported that reactive astrocytes produce abnormally large amounts of GABA due to aberrant monoamine oxidase B (MAOB) activity that results in excess GABA being released via Bestrophin-1 (Best-1) channels [34]. As GABA_C_ receptors are expressed on astrocytes [25,26,27], it is possible that inhibiting the production and therefore release of GABA from astrocytes via dampening GABA_C_ receptor activity could also contribute to a decrease in astrocyte reactivity, decrease in tonic inhibition, and increase in GAT3 expression and restoration of peri-infarct synaptic plasticity.

After stroke, astrocytes undergo many morphological changes to form a glial scar; these reactive astrocytes increasingly express GFAP, and this is used as a measure to determine the number and or size of the astrocytes. The glial scar has many roles after a stroke has occurred, both as a regulator of the inflammatory response and in relation to axonal sprouting, which is critical for cortical remapping [13,35,36,37]. The exact role that reactive astrocytes play after stroke still remains to be fully elucidated, although reactive astrocytes are known to be both beneficial and detrimental during different phases of stroke recovery [13,38]. For instance, the glial scar prevents damage from spreading throughout the brain; however, these reactive astrocytes also express a variety of neurite-growth inhibitors such as ephrin-A5, which limits functional recovery [39].

Here, we see a decrease in GFAP fluorescence after 5 mM (*R*)-4-ACPBPA, indicating a decrease in reactive astrocytes in this area. We hypothesize that the decrease of reactive astrocytes is beneficial, given that it correlates with functional recovery. This seems to be in contrast with the increase in GAT3 expression; however, after stroke, GAT3 expression shifts from being almost exclusively on astrocytes to neurons [40]. The mechanism through which (*R*)-4-ACPBPA is decreasing the number or size of the reactive astrocytes seems to be via GABA_C_
*ρ*2 receptor specific inhibition given that L655,708 was without effect.

Recommendations for preclinical stroke recovery studies demand different approaches than those for stroke neuroprotection, as stroke recovery studies are concerned with long-term recovery. For this, it is not ideal for stroke models to produce substantial animal death, such as what is observed using the filament occlusion middle cerebral artery models, and models should result in chronic behavioral deficits. These two simple criteria have guided consensus groups to make recommendations for using models that create focal lesions of similar size to those observed in humans [41]. The photothrombotic stroke model is reproducible, relatively non-invasive, and technically simple, and it is a model recommended by the stroke research recovery roundtable as being ideal for stroke recovery studies [41]. Disadvantages include a smaller rim of partially damaged peri-infarct cortex, simultaneous multivessel occlusion, and in some animals, the presence of edema, which is not commonly observed in human stroke. Still, the lesion can be positioned in any cortical location targeting particular deficits, and it is useful for assessing mechanistic and pharmacological studies of recovery.

## 5. Conclusions

In conclusion, the present study identified GABA_C_
*ρ*-subunits as possible targets for stroke recovery. We found that (*R*)-4-ACPBPA is more effective than (*S*)-4-ACPBPA at the doses tested, which is possibly because this compound has more affinity for the *ρ*2 containing receptors over other GABA_A_ receptor subtypes such as α5-containing receptors. Even though the motor recovery data indicates recovery that can be attributed to a change in tonic inhibitory currents, future studies should confirm this. In addition, as tonic inhibitory currents differ between males and females and with age, future studies should be undertaken to assess the efficacy of these compounds in these groups.

## Figures and Tables

**Figure 1 brainsci-11-00315-f001:**
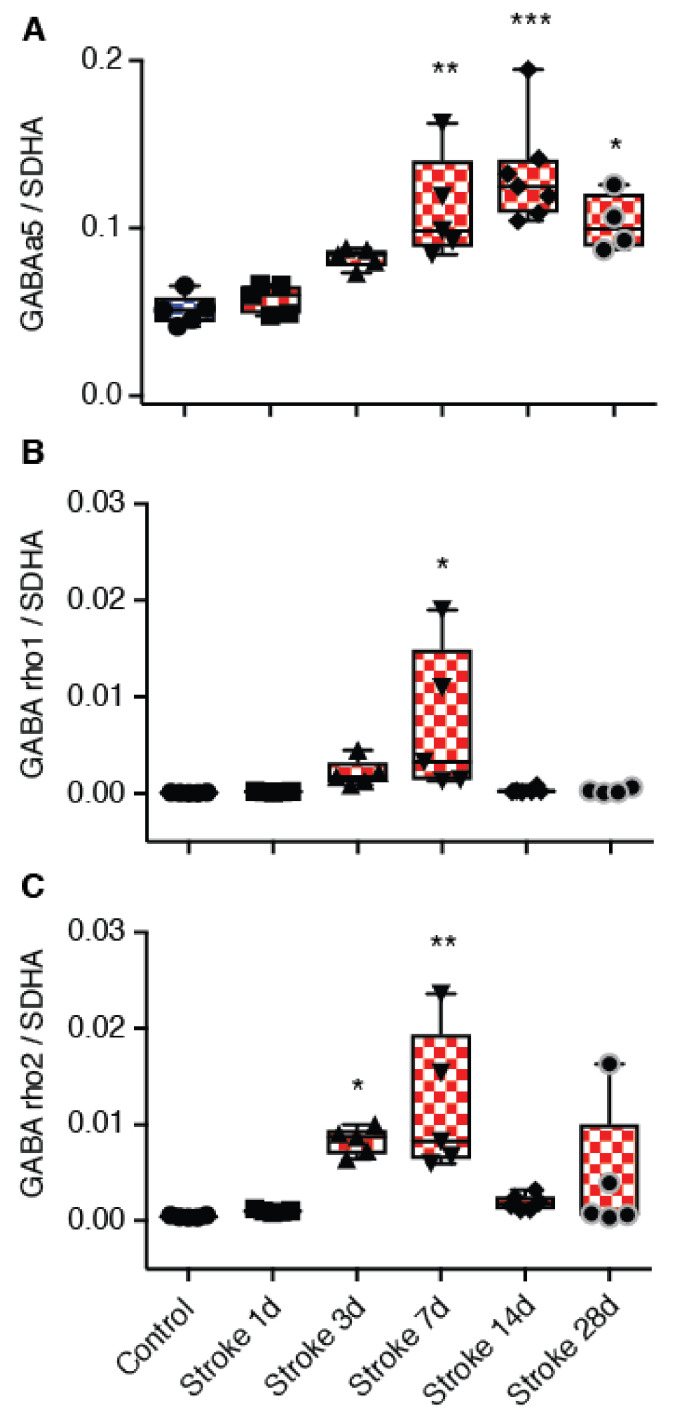
Temporal changes in α5, *ρ*1, and *ρ*2-subunit expression were assessed in the peri-infarct region using real time qPCR after photothrombotic stroke to the motor cortex. (**A**) GABA_A_ α5 expression, (**B**) GABA_C_
*ρ*1 expression, and (**C**) GABA_C_
*ρ*2 expression was assessed in stroke (red) and sham (blue) animals. * *p* < 0.05, ** *p* < 0.01, *** *p* < 0.001 (*n* = 5 for sham and 1, 3, 7, 14 and 28-days post-stroke). Data expressed as box plot (boxes, 25–75%; whiskers, minimum and maximum; lines, median). GABA: γ-aminobutyric acid.

**Figure 2 brainsci-11-00315-f002:**
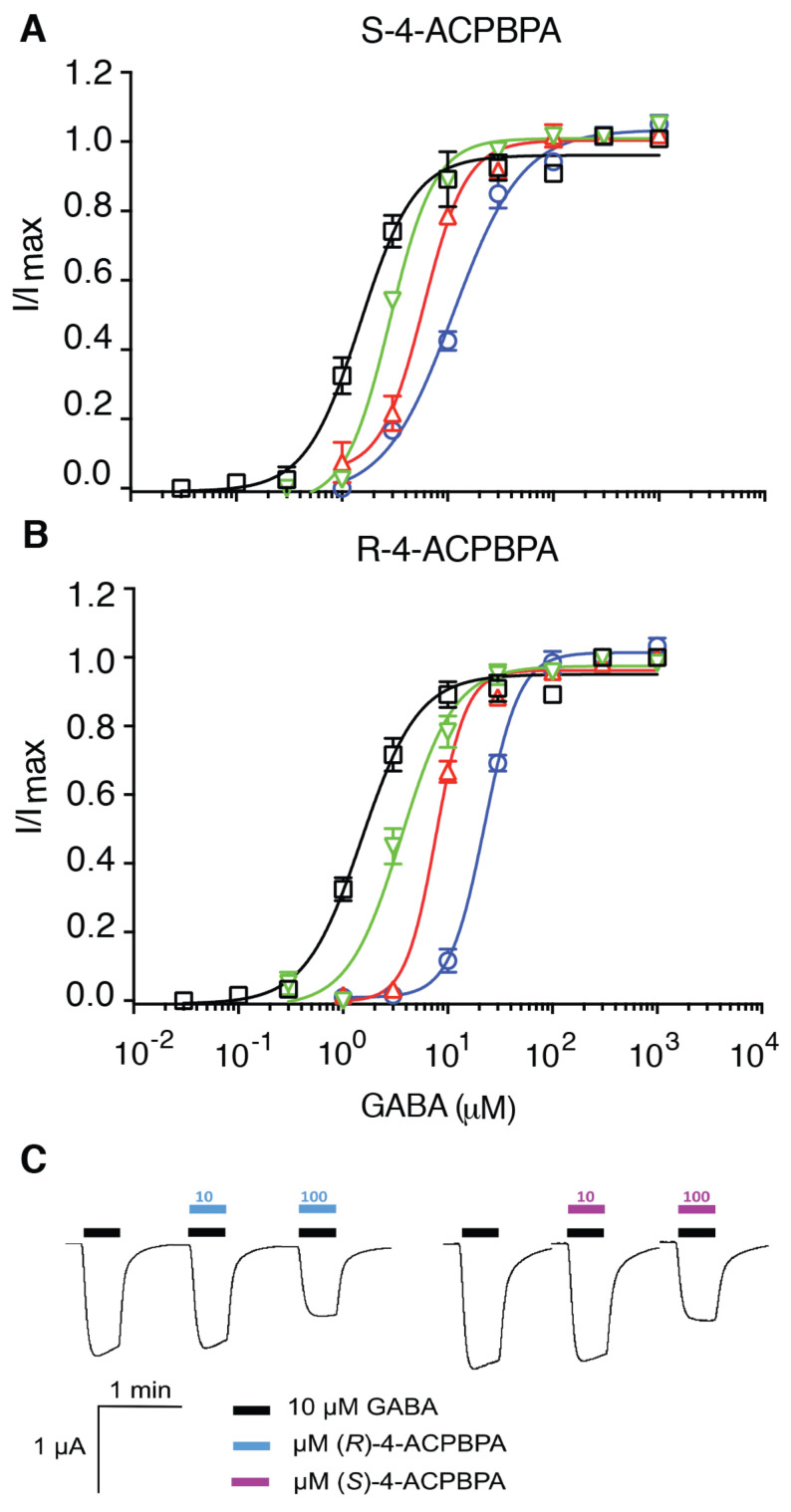
Dose–response curves of GABA (☐—open black boxes) in the presence of (**A**) 10 µM (▽—green triangle), 30 µM (△—red triangle), and 100 µM (◯—blue circle) (*S*)-4-ACPBPA and (**B**) 10 µM (▽—green triangle), 30 µM (△—red triangle), and 60 µM (◯—blue circle) (*R*)-4-ACPBPA at human recombinant *ρ*2 GABA_C_ receptors expressed in *xenopus* oocytes. (**C**) Representative responses of α5β2γ2 receptors expressed in xenopus oocytes to GABA (10 µM) alone and GABA (10 µM) with inhibitors (*R*) and (*S*)-4-ACPBPA (10 and 100 µM). (*R*) and (*S*)-4-ACPBPA (10 µM) inhibited GABA (10 µM) response by 2.5 ± 1.7 and 5.5 ± 2.5%, respectively, whilst (*R*) and (*S*)-4-ACPBPA (100 µM) inhibited GABA (10 µM) response by 36 ± 0.8 and 37 ± 0.6%, respectively. Data are expressed are mean ± SEM (*n* = 6 oocytes). (*R*)-4-ACPBPA: (*R*)-4-amino-cyclopent-1-enyl butylphosphinic acid, (*S*)-4-ACPBPA: (*S*)-4-amino-cyclopent-1-enyl butylphosphinic acid.

**Figure 3 brainsci-11-00315-f003:**
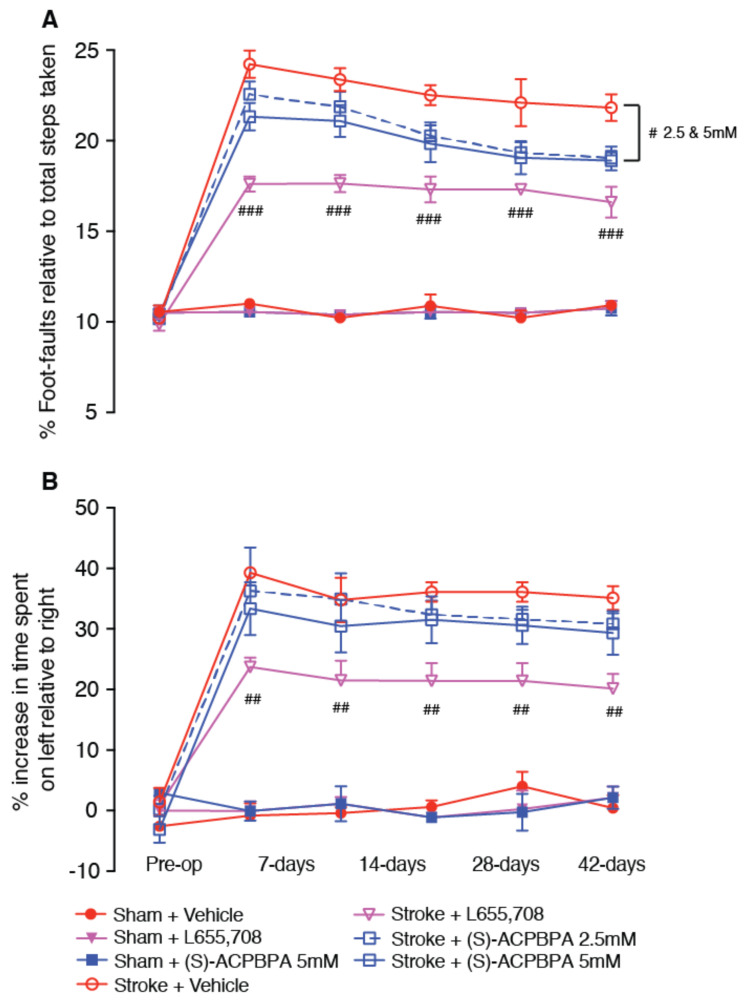
Behavioral recovery was assessed pre-op (before) and 7, 14, 21, 28, and 42 days after a photothrombotic stroke. NAM, 5 mM; L655,708, and the GABA_C_
*ρ*1 antagonist, 2.5 and 5 mM (*S*)-4-ACPBPA. Motor function was assessed by analyses of (A) foot-faults and (B) forelimb asymmetry in the grid-walking and cylinder tasks, respectively. # *p* < 0.05, ## *p* < 0.01, ### *p* < 0.001, compared to stroke + vehicle. Data are expressed as mean ± SD for *n* = 10/treatment group.

**Figure 4 brainsci-11-00315-f004:**
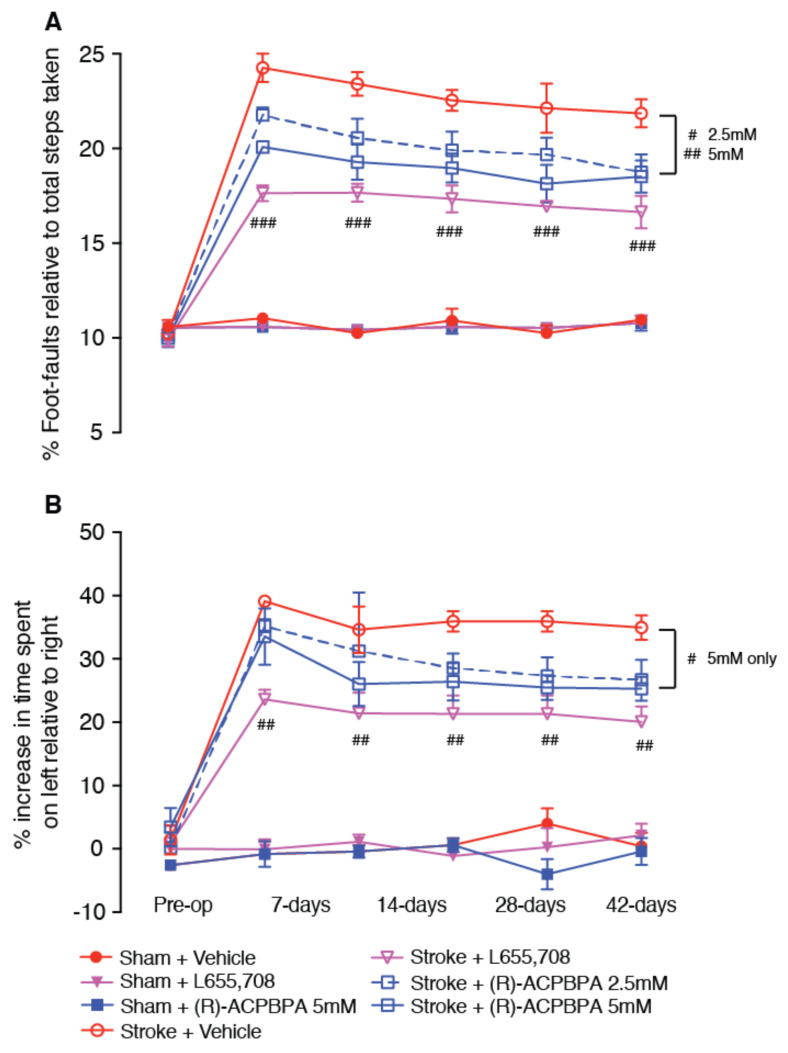
Behavioral recovery was assessed pre-op (before) and 7, 14, 21, 28, and 42 days after a photothrombotic stroke in the presence of the GABA_A_ α5 NAM, 5 mM L655,708, and the GABA_C_
*ρ*2 antagonist, 2.5 and 5 mM (R)-4-ACPBPA. Motor function was assessed by analyses of (**A**) foot-faults and (**B**) forelimb asymmetry in the grid-walking and cylinder tasks, respectively. # *p* < 0.05, ## *p* < 0.01, ### *p* < 0.001, compared to stroke + vehicle. Data are expressed as mean ± SD for *n* = 10 / treatment group.

**Figure 5 brainsci-11-00315-f005:**
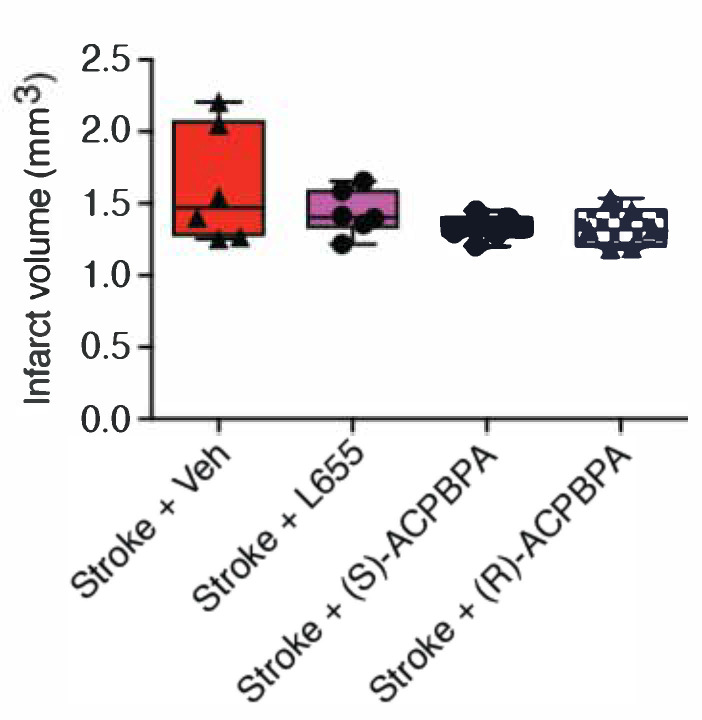
Assessment of infarct volume was carried out by quantifying cresyl violet stained sections generated 42 days post-stroke. Data expressed as box plot (boxes, 25–75%; whiskers, minimum and maximum; lines, median) for an *n* = 6/treatment group.

**Figure 6 brainsci-11-00315-f006:**
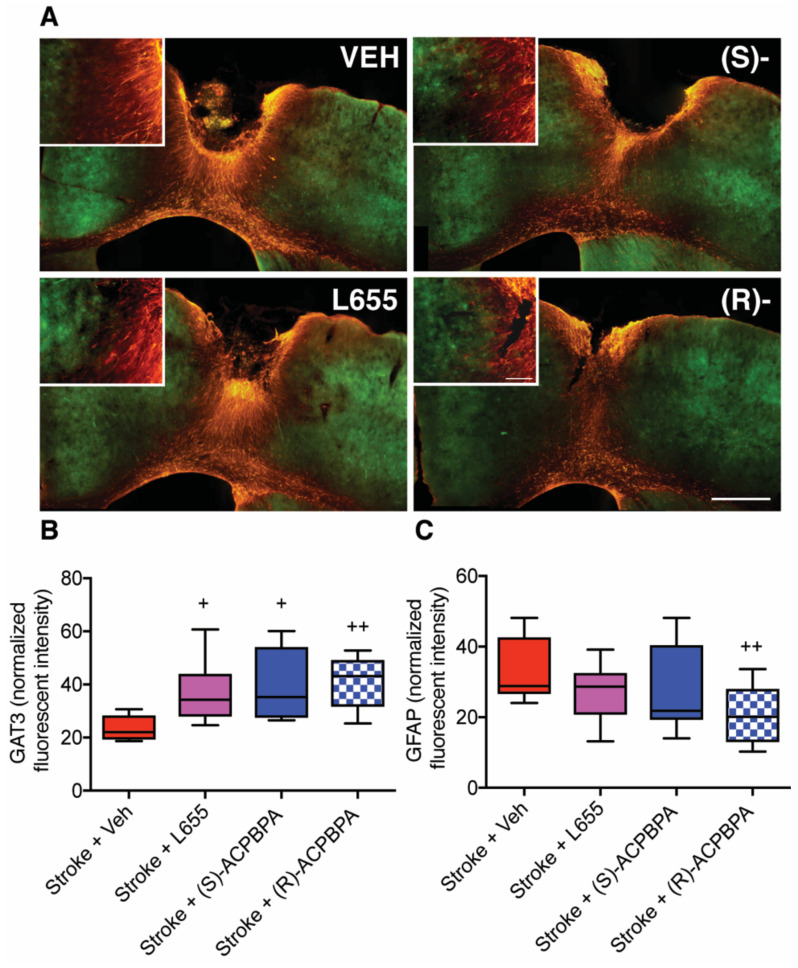
Changes in GABA transporter type 3 (GAT3) and glial fibrillary acidic protein (GFAP) expression were assessed 42 days post-stroke. (**A**) Representative photomicrographs showing GAT3 (green) and GFAP (red) labeling from stroke + vehicle (VEH), Stroke + L655,708 (L655), stroke + S-4-ACPBPA (S-), and stroke + (R)-4-ACPBPA (R-) High magnification images of each of the treatment groups from right next to the stroke border. Normalized fluorescent intensity measurements for GAT3 (**B**) and GFAP (**C**) were obtained from peri-infarct regions 400 µm from the stroke border. + *p* < 0.05, ++ *p* < 0.01. Data expressed as box plot (boxes, 25–75%; whiskers, minimum and maximum; lines, median) for an *n* = 6/treatment group. The scale bar in the insert = 100 um, whereas the scale bar shown in the main photomicrograph = 400 um.

**Figure 7 brainsci-11-00315-f007:**
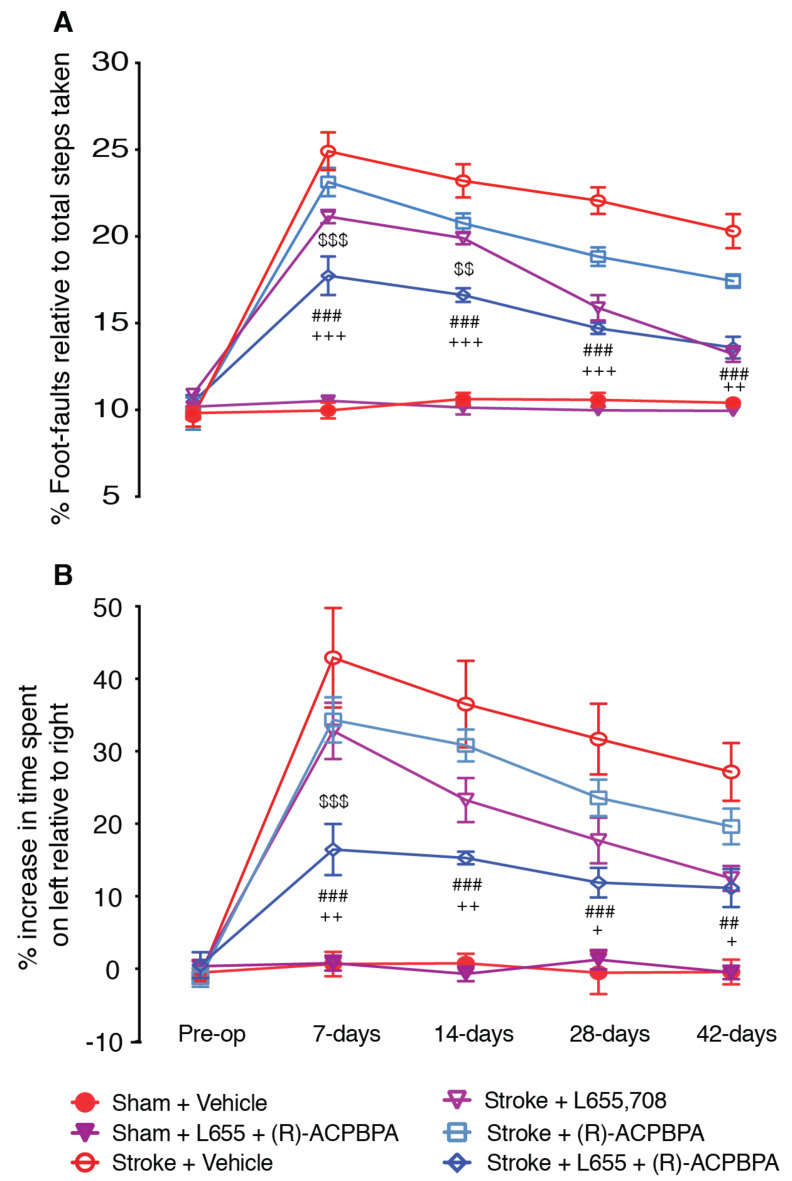
Behavioral recovery was assessed pre-op (before) and 7, 14, 21, 28, and 42 days after a photothrombotic stroke in the presence of the GABA_A_ α5 NAM, 5 mM L655,708, the GABA_C_
*ρ*2 antagonist, 5 mM (R)-4-ACPBPA, or a combined 5 mM L655,708 and 5 mM (R)-4-ACPBPA. Motor function was assessed by analyses of (**A**) foot-faults and (**B**) forelimb asymmetry in the grid-walking and cylinder tasks, respectively. ## *p* < 0.01, ### *p* < 0.001, compared to stroke + vehicle; + *p* < 0.05, ++ *p* < 0.01, +++ *p* < 0.001, compared to sham + L655,708 + (R)-4-ACPBPA; $$ *p* < 0.01, $$$ *p* < 0.001, compared to stroke + L655,708. Data are expressed as mean ± SD for *n* = 8 / treatment group.

## Data Availability

All data is stored on managed institutional servers and will be made avalibale upon request.

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
