# Peer review of "Targeting GABA_C_ Receptors Improves Post-Stroke Motor Recovery"

_brainsci, 2021, doi:10.3390/brainsci11030315_

Round 1

Reviewer 1 Report

The authors examined the effect of different GABA receptor modulators on stroke recovery. They show elevated levels of GABA-alpha5 and C receptor expression in peri-infarct cortex. Treatment of young adult mice with these drugs alone or in combination led to a significant improvement in sensori-motor function after stroke. I have provided some minor suggestions to improve the manuscript.

  • Methods: please provide more details about Imaging: objective, NA, x-y-z pixel sampling, camera, bit depth
  • Results: please report ANOVA statistics (F and df, Main effect of Treatment) for significant effects of treatment in behavioural, GFAP and GAT3 expression data
  • Figure 6: please show an additional higher mag image of GAT3/GFAP expression in peri-infarct cortex. Y axis or legend should state normalized fluorescent intensity
  • Did combined treatment alter infarct volume?

Author Response

The authors examined the effect of different GABA receptor modulators on stroke recovery. They show elevated levels of GABA-alpha5 and C receptor expression in peri-infarct cortex. Treatment of young adult mice with these drugs alone or in combination led to a significant improvement in sensori-motor function after stroke. I have provided some minor suggestions to improve the manuscript.

We thanks the reviewer for the kind words and careful critique of our manuscript.

  • Methods: please provide more details about Imaging: objective, NA, x-y-z pixel sampling, camera, bit depth

We have updated the methods to include this information:

“Imaging was undertaken on an Olympus BX61 microscope using a QImaging camera (Micropublisher 5.0 RTV, 2560×1920 pixels, 3.4µm pixel size) and 20× UPlanSApo objective (0.75NA).”

  • Results: please report ANOVA statistics (F and df, Main effect of Treatment) for significant effects of treatment in behavioural, GFAP and GAT3 expression data

We thank the reviewer for this comment and agree in part that this should be added.  Where treatment effects are great for instance the behavioural data, including the F-stats and df for Treatment effects, doesn't add anything to the paper.  However, where variance is not great, such as for the data on GFAP and GAT3 expression, then this adds value. For this we have added the ANOVA stats for the latter only.

  • Figure 6: please show an additional higher mag image of GAT3/GFAP expression in peri-infarct cortex. Y axis or legend should state normalized fluorescent intensity

We thank the reviewer for this comment.  We have added additional higher mag images to the top let corner of each image panel.  We have also added two scalebars, one on the new insert, which is 100um and one on the original images, which is 400um. We apologise for excluding the scalebars from the original submission. As suggested by the reviewer we have changed the y-axis to normalized fluorescent intensity.

  • Did combined treatment alter infarct volume?

No effect on infarct volume was observed.   

Reviewer 2 Report

The aim of the present study was to test the hypothesis that GABAc receptor contributes to the excess of tonic GABA inhibition of the peri-infarcted cortex during post-stroke recovery. This pre-clinical analysis is original, and the data are very interesting. It shows for the first time that the source of tonic inhibition around a stroke lesion is not only due to tonic GABAA Receptor. Very interestingly the author shows a progressive increase in the expression of GABAc receptors. Although the blockade of GABAcR is not as efficient as the blockades of extrasynaptic alpha5-GABAAR, the full inhibition (GABA alpha 5-R + GABAcR) is remarkably efficient to boost recovery. The current manuscript is well designed and written. The data support the conclusions. Addressing the following concerns would improve the study.

  1. In the rational for the choice of Alzet pump administration, was there evidence that the compounds actually cross the blood brain barrier?
  2. The calculation method of the infarct size should be reported in the M&M section.
  3. Discussion: "It is thought that the increase in tonic inhibition is an intrinsic mechanism that helps limit the spread of damage in the brain and silences communication so the brain can repair itself." The authors should cite a reference or data supporting this.
  4. Discussion: what could be the putative mechanism of GAT-3 modulation by extrasynaptic GABAcR blockers?
  5. Legend of Figure 6: change “the stroke boarder” by border.
  6. Figure 6: a scale bar is missing. Also, what ROI was use for the quantification?
  7. Homogenize behavior vs. behaviour.
  8. Verify Ref #1, #21 for volume and pages.

Author Response

The aim of the present study was to test the hypothesis that GABAc receptor contributes to the excess of tonic GABA inhibition of the peri-infarcted cortex during post-stroke recovery. This pre-clinical analysis is original, and the data are very interesting. It shows for the first time that the source of tonic inhibition around a stroke lesion is not only due to tonic GABAA Receptor. Very interestingly the author shows a progressive increase in the expression of GABAc receptors. Although the blockade of GABAcR is not as efficient as the blockades of extrasynaptic alpha5-GABAAR, the full inhibition (GABA alpha 5-R + GABAcR) is remarkably efficient to boost recovery. The current manuscript is well designed and written. The data support the conclusions. Addressing the following concerns would improve the study.

We thanks the reviewer for the kind words and careful critique of our manuscript.

  1. In the rational for the choice of Alzet pump administration, was there evidence that the compounds actually cross the blood brain barrier?

Yes, all compounds cross the BBB and we have also used minipumps in the past for administering GABA compounds such as L655,708.  We have added the following to the paper to address this:

All compounds have previously been tested in vivo and shown to cross the BBB and have an effect on the brain, including L655,708, which we have previously reported is effective after stroke [5].

  1. The calculation method of the infarct size should be reported in the M&M section.

The following text and formula has been added to the methods section:

“Images of cresyl violet staining were taken using an inverted montaging microscope (Model Ti2E Wideview, Nikon, Japan) set with a 2.5x objective lens. Images were then exported as TIFF files and opened on Fiji ImageJ to quantify infarct volume. Stroke volume was calculated as per the equation:”

  1. Discussion: "It is thought that the increase in tonic inhibition is an intrinsic mechanism that helps limit the spread of damage in the brain and silences communication so the brain can repair itself." The authors should cite a reference or data supporting this.

We have added the following two references to support this statement. Refs 5 and 17. Other references have been checked and order has been updated where needed based on changes.

Clarkson, A.N.; Huang, B.S.; Macisaac, S.E.; Mody, I.; Carmichael, S.T. Reducing excessive gaba-mediated tonic inhibition promotes functional recovery after stroke. Nature 2010, 468, 305-309.

Clarkson, A.N.; Boothman-Burrell, L.; Dosa, Z.; Nagaraja, R.Y.; Jin, L.; Parker, K.; van Nieuwenhuijzen, P.S.; Neumann, S.; Gowing, E.K.; Gavande, N., et al. The flavonoid, 2'-methoxy-6-methylflavone, affords neuroprotection following focal cerebral ischaemia. Journal of cerebral blood flow and metabolism : official journal of the International Society of Cerebral Blood Flow and Metabolism 2019, 39, 1266-1282.

  1. Discussion: what could be the putative mechanism of GAT-3 modulation by extrasynaptic GABAcR blockers?

This is an interesting question.  We have added the following to the discussions section to addess this point

“In addition to the GAT3 mediated increases in tonic inhibitory currents, previous research has also reported that reactive astrocytes produce abnormally large amounts of GABA due to aberrant monoamine oxidase B (MAOB) activity that results in excess GABA being released via Bestrophin-1 (Best-1) channels [34]. As GABAC receptors are expressed on astrocytes [25-27], it is possible that inhibiting the production and therefore release of GABA from astrocytes via dampening GABAC receptor acitivity could also contribute to a decrease in astrocyte reactivity, decrease in tonic inhibition, and increase in GAT3 expression and restoration of peri-infarct synaptic plasticity.”

Reference 34 (new): Jo S, Yarishkin O, Hwang YJ, Chun YE, Park M, Woo DH, Bae JY, Kim T, Lee J, Chun H, Park HJ, Lee DY, Hong J, Kim HY, Oh SJ, Park SJ, Lee H, Yoon BE, Kim Y, Jeong Y, Shim I, Bae YC, Cho J, Kowall NW, Ryu H, Hwang E, Kim D, Lee CJ. GABA from reactive astrocytes impairs memory in mouse models of Alzheimer's disease. Nat Med. 2014, 20, 886-96.

  1. Legend of Figure 6: change “the stroke boarder” by border.

We have corrected this error.

  1. Figure 6: a scale bar is missing. Also, what ROI was use for the quantification?

Please see response to review 1 as well regarding figure 6.  We have now added in a scalebars to this figure. 

  1. Homogenize behavior vs. behaviour.

We have carefully read back over our paper and corrected the UK to US spelling. We thank the reviewer for picking this up and apologise for this oversight.

  1. Verify Ref #1, #21 for volume and pages.

We have updated these references.
